# One-shot phase-recovery using a cellphone RGB camera on a Jamin-Lebedeff microscope

**Benedict Diederich**[1,2]☯*, **Barbora Marsikova**[1,2]☯, **Brad Amos**[3], **Rainer Heintzmann**[1,2]

**1** Leibniz Institute of Photonic Technology, Albert-Einstein-Straße 9, 07745 Jena, Germany, **2** Institute of Physical Chemistry and Abbe Center of Photonics, Friedrich-Schiller-University, Helmholtzweg 4, 07745 Jena, Germany, **3** Medical Research Council, MRC, Laboratory of Molecular Biology, Cambridge, United Kingdom

☯ These authors contributed equally to this work.
\* benedict.diederich@ipht-jena.de

**Data Availability Statement:** All data and image processing algorithms can be found in the github repository: https://github.com/beniroquai/Tensorflow_Jamin-Lebedeff.

## Abstract

Jamin-Lebedeff (JL) polarization interference microscopy is a classical method for determining the change in the optical path of transparent tissues. Whilst a differential interference contrast (DIC) microscopy interferes an image with itself shifted by half a point spread function, the shear between the object and reference image in a JL-microscope is about half the field of view. The optical path difference (OPD) between the sample and reference region (assumed to be empty) is encoded into a color by white-light interference. From a color-table, the Michel-Levy chart, the OPD can be deduced. In cytology JL-imaging can be used as a way to determine the OPD which closely corresponds to the dry mass per area of cells in a single image. Like in other interference microscopy methods (e.g. holography), we present a phase retrieval method relying on single-shot measurements only, thus allowing real-time quantitative phase measurements. This is achieved by adding several customized 3D-printed parts (e.g. rotational polarization-filter holders) and a modern cellphone with an RGB-camera to the Jamin-Lebedeff setup, thus bringing an old microscope back to life. The algorithm is calibrated using a reference image of a known phase object (e.g. optical fiber). A gradient-descent based inverse problem generates an inverse look-up-table (LUT) which is used to convert the measured RGB signal of a phase-sample into an OPD. To account for possible ambiguities in the phase-map or phase-unwrapping artifacts we introduce a total-variation based regularization. We present results from fixed and living biological samples as well as reference samples for comparison.

## Introduction

Microscopy is a powerful tool not only for investigation of the micro-structure of an object but also for quantification of local properties of the sample. One such property is the optical path difference (OPD) of the light passing through biological matter, which is known to be closely related to the dry mass per area density of biological matter [1, 2].

**Funding:** This project was supported by the free state of Thuringia and the DAAD GSSP program for studying abroad.

**Competing interests:** The authors have declared that no competing interests exist.

The classical brightfield-microscope, well established for almost two centuries [3, 4], is mostly limited to visualize amplitude-contrast e.g. absorptive structures of a sample. By adding optical components altering the phase of the light, Fritz Zernike created the well-known Phase contrast [5] technique greatly enhancing the visibility of weakly absorbing objects such as single cells, yet lacking quantification. Later methods like the Differential Interference Contrast microscopy (DIC) followed this idea of converting a pure phase-variation in the object into amplitude contrast by the interference nature of light [5]. By taking a series of DIC images under different shearing orientation, quantitative phase information can be recovered, a method termed quantitative DIC [6, 7].

Various ways exist to recover quantitative phase information based on some form sequential image acquisition exploiting phase shifting [8] or change of the illumination conditions [9–11]. However to gain such quantitative phase information from a single image, one either needs to do off-axis holography [12, 13] or simultaneously acquire multiple phase images [14–16]. Yet in white-light interferometry it is common to acquire a multi-color image to recover the phase information. This method is closely related to the working principle of the JL-microscope, as detailed further below. If target-specific molecular information is needed, fluorescent labels are the method of choice, as they can be specifically attached to the molecule of interest in the biological sample. The properties of such labels can also be exploited to circumvent the optical diffraction limit formulated by Abbe in 1851 [3]; in techniques such as Structured Illumination Microscopy (SIM) [17], (direct) Stochastic Optical Reconstruction Microscopy (*d*STORM) [18] or stimulated emission microscopy [19]. However, labelling a sample means interfering with its biology and the required strong illumination in fluorescence imaging can also be harmful to living organisms [20].

Recent advances in computational imaging combined ordinary microscopes with tailored algorithms in order to obtain sample information without the need for labelling. Quantitative phase measurements using the Jamin-Lebedeff microscope never found its wide application, which is mostly due to the rarity of the equipment and more importantly, because of the lack of an easy-to-use automated quantification process (e.g. single-shot measurements). In general, there are two different methods to recover the OPD in JL-microscopy.

The phase-shifting technique records a set of images using monochromatic light (e.g. $\lambda_c = 546$ *nm*) which passes a rotating analyzer and a $\lambda/4$ plate (Senarmont Compensator). The resulting Senarmont measurement (i.e. varying interference stripes), gives precise information about the local OPD. Alternatively, a single image at polychromatic light is acquired, where the observer compares the colors to a reference color-table (Michel-Levy chart) to estimate the OPD.

To our knowledge no image processing procedure has yet been developed to automatically acquire quantitative data using a LJ-microscope. Yet for the visualization of even small phase differences, this technique is rather outstanding, as it promotes an exquisite sensitivity [21].

In our study we attempt to bring an old Jamin-Lebedeff microscope (Carl Zeiss West, Confocal) back-to-live using a set of 3D-printed parts and the RGB-camera of an ordinary cellphone. Additionally, we present a novel technique to quantitate the phase using a gradient-based phase-retrieval algorithm which uses total variation (TV) regularization implemented using the open-source auto-differentiation toolbox Tensorflow [22].

This label-free technique relies on capturing a single reference image of a known object and does not need any additional modelling parameters. It can then produce phase-measurements in real-time because the color-encoded optical-path difference (OPD) can be recovered in a post-processing step.

Below we describe a new method which measures the OPD using the JL microscope combined with a cellphone in one shot.

## Materials and methods

### Setup and theory of image formation in Jamin-Lebedeff microscopy

The Jamin-Lebedeff Microscope encodes optical path difference into color [23]. The optical setup including the polarization states for the light in each plane is visualized in Fig 1. It works by first polarizing an incoherent light-source (e.g. a halogen lamp or LEDs) using a linear polarizer, before it passes through a specific condenser containing a birefringent beam-splitter. The two mutually coherent orthogonal linear polarization components are split and displaced with respect to each other before illuminating the sample. The light exiting the sample impinges on a carefully matched beam-combiner representing an identical birefringent calcite crystal plate, cut obliquely to its crystallographic axis. The principal directions are perpendicular to each other and form and angle of 45° with that of the polarizer. For better understanding of the mutual polarization axes, see Fig 1. The beam splitter is mounted directly above the condenser lens and splits the light into two rays vibrating perpendicularly to each other. The ordinary beam forms the laterally (and axially) displaced reference beam, whereas the extraordinary beam is the object beam [24]. The beam-combiner in front of the objective lens generates interference, the previously ordinary beam must now become its extraordinary beam and the extraordinary one must become ordinary. This is achieved by a λ/2 plate which rotates the vibration plane by 90° right after leaving the beam splitter [25].

To observe interference, the path length of both beams must be close to equal. This is achieved, by fist adjusting the system carefully on an empty sample region. However, very good compensation is achieved only for specified wavelength (e.g. λ = 546 *nm*). Linearly Polarized light which does not meet this condition is converted into elliptically polarized light. The beam combiner is mounted in front of the objective lens. It recombines the measuring and reference beam. It has to be carefully aligned for best contrast. The sample is inserted between the λ/2 plate and the beam combiner. If there is no object in the beam path, both beams are

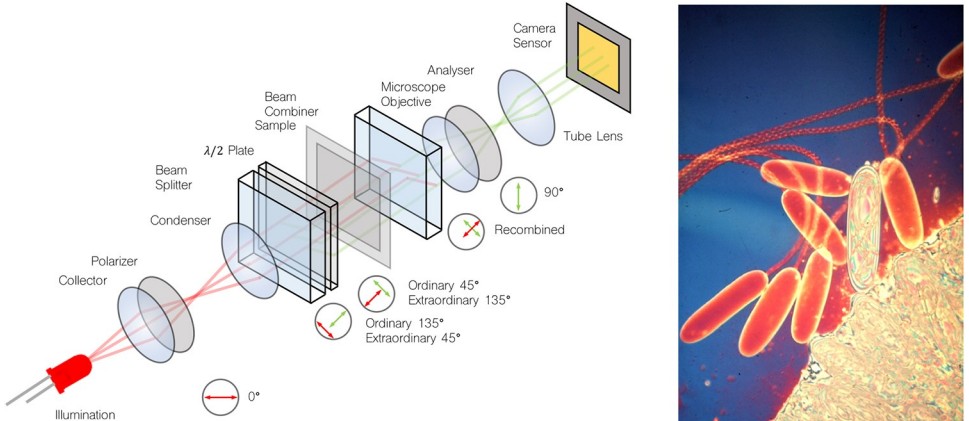

**Fig 1. Optical setup Jamin Lebedeff microscope.** Optical setup of an Jamin-Lebedef interference microscope which follows Köhler illumination. A collimated white-light source is filtered by a polarizer before a combination of a λ/2 -plate and a beam-splitter splits the ordinary (45°) and extraordinary (135°) light-path by roughly half the field of view of the illuminated sample region. The transparent object adds a locally varying phase-factor to the sample beam, whereas the reference beam is unaltered. An adjacent beam-combiner joins the two light-paths before the objective lens images the interference onto a sensor. A ghost-image appears only for those wavelength where the λ/2 -plate does not cancel out the constructive/destructive interference entirely (i.e. white-light). The image on the right shows seven nematocysts near the surface of a tentacle of the anemone Stoichactis. Six are empty capsules which have exploded, but one shows the internal thread not yet everted during explosion. The empty capsules show red/orange first order interference colors; the unexploded capsule the white and other high-order colors due to its high dry mass per unit area. (Lubbock & Amos, (1981) Nature 290, 500-501).

identical and both polarizations vibrate in phase and a uniform dark background is observed. In case the sample beam passes through a medium with spatially varying refractive index whereas the reference (p-polarized) passes through an empty region, the OPD generates a phase difference $\Delta\alpha$ given by:

$$\Delta\alpha = \pi/\lambda \cdot d \cdot (n_R - n_O), \tag{1}$$

$$OPD = d \cdot (n_R - n_O), \tag{2}$$

with $n_R$ representing the RI of the reference material, $n_O$ the RI of the object. $\lambda$ is given by the centre wavelength of the illumination and $d$ gives the thickness of the sample along the optical axis.

When passing through the analyser the field-components of polarized light vibrating in its absorption direction are absorbed, while the components vibrating in its transmission direction are transmitted and form an interference pattern right after the analyser.

The above-mentioned components are part of a Carl-Zeiss Jamin-Lebedeff transmitted light interference equipment which was designed to use for the Zeiss STANDART POL-microscope (Zeiss West). We adapted the specialized LJ-optics to a Zeiss West Confocal microscope body (used to be part of a confocal microscope) using customized 3D printed components Fig 2. A rotatable mechanism for the polarizer and analyser allows for setting the correct mutual position of these filters, which is crucial for the interference contrast. All parts can be downloaded from [26].

## Interference: From wavelength to Color

Illuminating with monochromatic light, a pattern of bright and dark fringes is visible corresponding to constructive and destructive interference at a phase object [26]. In case of polychromatic (white) light the entire spectrum contributes to the interference causing a dependence of the observed color on the OPD (see Fig 1 right). The resulting interference

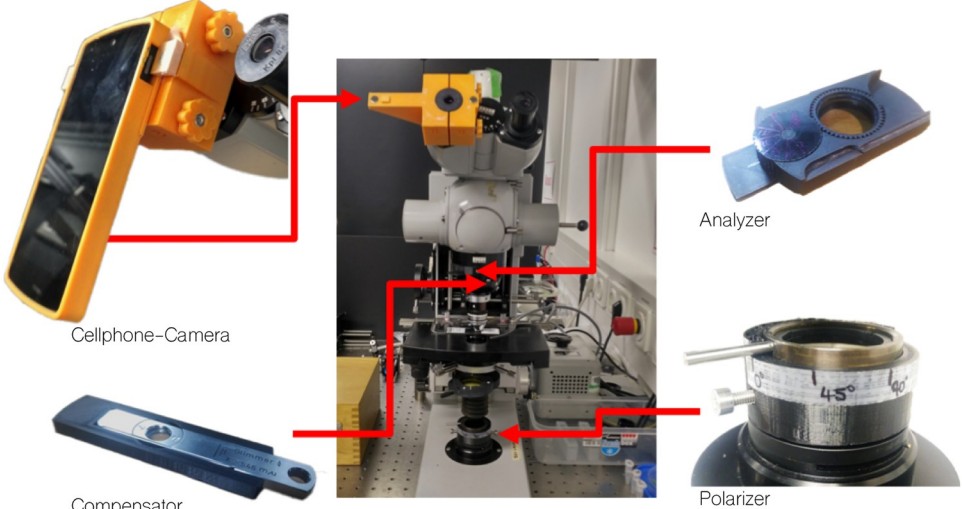

**Fig 2. 3D-printed components to update the Jamin-Lebedeff.** A set of different 3D printed adapters to get the old microscope to work with the Jamin-Lebedeff components. A customized smartphone mount using a magnetic-fit aligns the cellphone lens to the optical axis of the eye-piece. The analyser is equipped with a rotating mechanism to tune the orientation of the polarization filter. The compensator and polarizer got 3D printed adapters to fit into the confocal microscope.

**Table 1. Table of different microscope objective lenses available for the JL system.**

|  | 10×, NA = 0.22 | 30×, NA = 0.65 | 100×, NA = 1.0 Oil |
|---|---|---|---|
| Shear | 500 $\mu m$ | 170 $\mu m$ | 50 $\mu m$ |
| FOV | 1400 $\mu m$ | 335 $\mu m$ | 145 $\mu m$ |
| Ghost Image | 565 $\mu m$ | 187 $\mu m$ | 55 $\mu m$ |

colors are classified in orders conveniently described by the Michel-Levy chart. Although this chart was originally made for measuring birefringence and determination of birefringent materials, it also conveniently visualizes the colors observed using the Jamin-Lebedeff microscope. When setting up the microscope, it is advisable to set the background to black/dark purple. Each phase variation other than the background-medium starts around the first order of white-light interference and therefore shows highest contrast (i.e. less ambiguous in its color-response). The observed color also depends on the spectrum of the light-source and the filter characteristics and potentially the image processing of the imaging device (e.g. the Bayer-pattern of the camera).

When a sample is observed, a darker foggy ghost image is seen. This is caused by the aberrated (i.e. astigmatic) part of the reference beam [27], which also passes through the object, since the beam is largely laterally shifted in the beam combiner. The amount of relative shear between the reference and the object beam (field of view and ghost image shift) depending on the objective used, is listed in Table 1. This ghost image makes imaging of dense objects impossible as false colors caused by superposing of the real and the shifted ghost image produce incorrect results.

The resolution, using the 40× objective, goes beyond 228.1 line-pairs/mm of the USAF resolution target.

## Imaging using a cellphone

Recent studies have shown that overall camera performance (e.g. noise, sampling rate etc.) makes replacing expensive sCMOS/emCCD sensors by industry-grade CMOS [28] or even cellphone cameras [29] possible. Especially when it comes to RGB imaging, cellphone cameras are omnipresent. In our study we have tested the Nexus 5 (LG/Google, Japan) and the Huawei P9 pro (China). We connected both of them to a standard binocular-eyepiece (10×, Carl-Zeiss) using a customized 3D-printed Adapter (Fig 2). Correct imaging is achieved if the exit pupil of the microscope matches the entrance pupil of the cellphone's camera lens.

While measuring the chromatic response of the RGB-sensor of the LG Nexus 5 (Sony Exmor IMX179, 8.0 MP, pixel pitch = 1.4$\mu m$), we recognized, that the signal in the blue-channel is very weak and leads to increased noise in our experiments. This causes a noisy pixel-response with close overlaps in the Look-Up-Table (LUT) making the phase recovery more difficult as further discussed in the following section.

Relying on the newer Huawei P9, equipped with a Bayer-patterned, back-illuminated, CMOS sensor (Sony Exmor IMX286, 12.0 MP, pixel pitch = 1.12$\mu m$) in combination with a digital controllable LED source (CooLED, pE4000) improve the SNR of the measurements by increasing the blue component in the light-source spectrum.

Compared to common RGB industry-grade cameras, the advantage of using cellphones is twofold. First, recent developments brought up high-sensitivity RGB cameras, which are not only widely spread, but also very compact which makes additional hardware-drivers or cables unnecessary. Secondly, the processing (e.g. phase-recovery) could be carried out on the mobile device itself, thus reducing additional complication of setting up software on a computer.

Throughout our experiments we used the open-source cellphone camera APP FreeDCam [30] to keep settings like white-balance, adaptive contrast compensation etc. constant. We acquired the frames in RAW-format to avoid any additional modifications of the data like JPEG-compression artefacts, auto-whitebalancing (AWB) or de-Bayering of the color-channels. In case of time-lapse or live-view imaging, we used the video mode of the camera APP FreeDCam with highest possible video-bitrate to reduce any possible compression artefacts. Additionally we locked the color-map and exposure settings between measurements and calibrations to avoid any alteration of the color response of the reference and actual sample.

## Image processing

Starting with the RGB color-coded microscopy images (see Fig 3) to calculate the gray-valued OPD maps corresponds to an inversion of a color-map with a many-to-one mapping instead of the common one-to-many mapping. Although there exists an analytical formula which relates RGB values to a given optical phase, this is heavily altered if experimental conditions, such as variations in the light-source spectrum, incorrect microscope settings or tone mappings from the cellphone's camera influence the RGB image. Therefore, we decided to estimate the object's phase by calibrating with a known phase object (e.g.a glass fiber) which is used to generate a LUT for new samples acquired under the same experimental conditions. This requires a-priori knowledge of the reference sample which can be calibrated once using third-party instruments.

The basic idea is to find a relationship between a given RGB-value and its corresponding OPD-value. In order to keep the influence of pixel-noise and dust in the microscope's optical path low we chose an optical fiber as a known reference object. This enabled to average over a larger number of RGB pixel or similar OPD-values. In order to build the LUT, the first step is

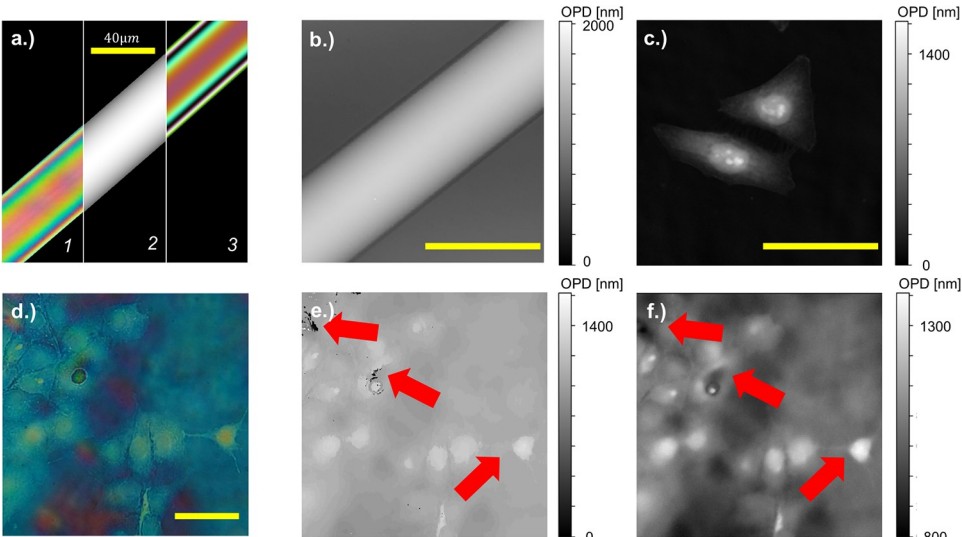

**Fig 3. Calibration and sample phase objects for the JL setup.** The calibration routine in a) requires a known phase-sample (e.g. optical fiber, (1)/left) with a phase-profile ((2)/middle) over several wavelengths. The forward model with a known LUT is given in the right (3) and roughly matches the measured RGB image using the cellphone (left). Additionally the optical fiber b) and the HeLa cells c) are imaged using a coherence controlled holographic microscope (CCHM, Q-Phase, Tescan and TELIGHT, Brno, Czech Republic) to have a ground-truth reference. A raw JL image captured with the Huawei P9 RGB camera is shown in d), where we computed the OPD using the minimal norm solution e) and the regularized iterative algorithm f). The iterative algorithm has less ambiguities and preserves fine structures better (indicated with red arrows).

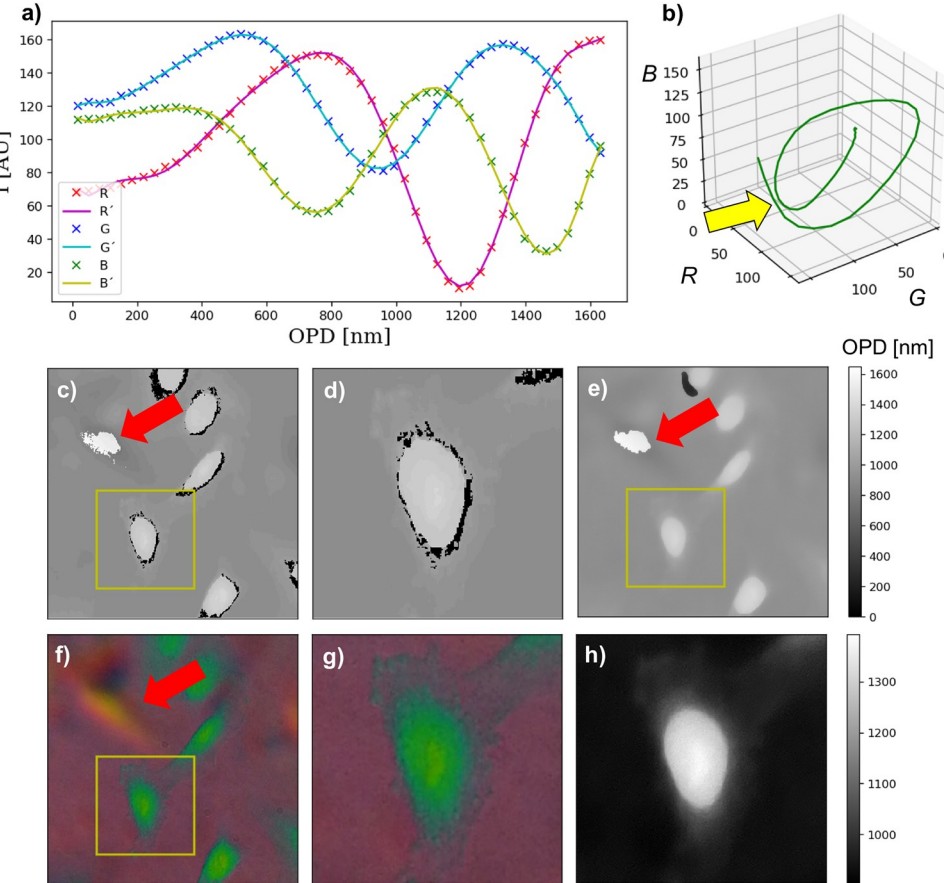

**Fig 4. Parametric LUT curve and reconstruction results.** The graphs in a) show the relationship between the reference phase-sample and the resulting color image for each RGB-channel individually as dotted points; dashed line shows the parametrized polynomial fit; b) displays this relationship as a 3D parametric plot. The red arrow indicates points where the curve is very close or even intersecting; here the algorithm has problems to distinguish between different OPDs with close RGB-values. The exemplary JL raw-image from fixed HeLa-cells is reconstructed using the minimal norm-solution in c)/d) and the iterative gradient-descent based algorithm in e)/h). The gradient-descent based algorithm removes the phase-ambiguities and preserves the fine features of the cell membrane compared in the zoomed-in version d) and h). In areas where ghost-images are very dominant (orange area in f)), both algorithms fail (red arrows).

to find the cylindrical shape in the RGB interference image. The LUT is created by averaging the pixel-values of all RGB pixels within a certain OPD-range. To reduce ambiguities, it is helpful to cluster a given OPD range of, in our case, $3 \cdot \lambda$ by dividing this range into e.g. 50 OPD bins. The result of this procedure is a parametric curve in 3D-space (see Fig 4b)) of a relation which is in general not invertible. The MATLAB-code (MATLAB 2017b, The Math-Works, Natick, USA.) can be downloaded from the github-repository [26].

The general idea of retrieving the OPD is now to find the closest RGB value within the LUT for a given pixel in the experimental dataset. Mathematically this can be expressed by minimizing the L2 distance:

$$argmin_{opd}((R[opd] - R_{meas})^2 + (G[opd] - G_{meas})^2 + (B[opd] - B_{meas})^2), \qquad (3)$$

where $R[opd]$ corresponds to one red color-value with integer index $opd \subset [opdminmin,$

*opdmax*]. $R_{meas}$ equals the measured red value of a single pixel. *G* and *B* correspond to the green and blue channel. Due to its discrete nature, this problem cannot robustly be solved using standard gradient-descent algorithms. Nevertheless, the solution to the above problem can quickly be found, by taking the OPD-index which minimizes the L2 norm for each pixel—also called the minimum norm solution.

Looking at the exemplary 3D curve in Fig 4b) which corresponds to the RGB-LUT, it can be seen that at certain points in space, the curve nearly intersects with itself (indicated by a yellow arrow). A pixel with RGB-values near such a point of intersection (yellow arrow) can easily be assigned to a wrong OPD, if affected by dust or noise or other small errors. This is because the algorithm is fitting the OPD individually pixel by pixel from the measured RGB data without considering relationships between surrounding pixels.

In order to overcome this noise problem, we optimized the illumination and added a regularization term to the above optimization problem which penalizes jumps in OPD between neighbouring pixels. This is conveniently achieved by taking an isotropic total-variation regularizer [31, 32].

The first step is to make the LUT-based problem differentiable. This is achieved by fitting an $n^{th}$-order polynomial into each color-channel as shown in Fig 4. We limited the order of the polynomial to 10. The fitting process also smooths the curve and eliminates additional noise as seen in the 3D parametric curve (Fig 4, orange). The optimization problem then reduces to finding a solution to the following equation:

$$argmin_{opd}[(R(opd) - R_{meas})^2 + (G(opd) - G_{meas})^2 + (B(opd) - B_{meas})^2 + R_{TV}(opd)], \quad (4)$$

where

$$R_{TV}(opd) = \sqrt{\left(\frac{\partial opd}{\partial dx}\right)^2 + \left(\frac{\partial opd}{\partial dx}\right)^2 + eps_0^2}. \quad (5)$$

$\lambda_{TV}$ is an empirically chosen weight for the influence of the regularization term, $eps_C$ makes the solution differentiable at the zero-position and controls the smoothness of the solution. Intuitively, the algorithm recovers the OPD in those areas where the phase-gradient is non-smooth e.g. at phase-jumps likely due to overlaps in the 3D parametric curve.

## Calibrating the microscope

Setting up the microscope can be done by observing the interference response on the cellphone's LCD screen. In an ideal case, the polarizer and analyser are set the way, so that the background (e.g. area without any object) is close to black. This means that any phase derivation from the background is interfering around the 0-th order. Throughout our experiments we relied on the 40×, NA = 0.65 objective lens with the matching condenser (Zeiss West Condenser Pol Int. II) with a variable numerical aperture which was set to realize Köhler illumination.

## Cell culture

HeLa cells (ATCC CCL-2) cells were routinely cultured in ATCC-formulated Eagle's Minimum Essential Medium (No. 30-2003) supplemented with 10.0% fetal bovine serum (No 30-2020) and 1% Penicillin-Streptomycin at 37.0˚C in a humidified atmosphere with 5.0% $CO_2$. Fetal bovine serum to a final concentration of 10. The rotifer was collected by filtering pond water through a coffee filter and diluting it in condensation water.

## Results

Based on the two different proposed image processing algorithms we want to retrieve the phase-information of the color-coded interference images from a JL microscope. Therefore, we have reanimated this 50-year old microscope by incorporating current date technology such as cellphone cameras and 3D printed adapters.

Fig 4 shows an exemplary result of a raw RGB measurement of fixed HeLa cells, where we applied the direct L2 pixel-wise and TV-regularized optimization algorithm to compute the quantitative result. Both algorithms need to be provided with a calibration measurement using a reference sample with known Phase-relation. Here we used a telecom glass-fiber (Thorlabs SMF28) with 8.2/125 $\mu m$ diameter of core and cladding that shrinks accordingly during tapering, (tapered to diameter 38 $\mu m$, n = 1,4585). This fiber is embedded in an immersion media (Mixture of glycerol and water, n = 1,4140), thus following in a maximum OPD of $3\lambda_0$ with $\lambda_0$ = 546 $nm$ which also limits to the maximum OPD-difference measurable with this configuration. As stated above, the fiber offers an OPD of $3\lambda_0$ which captures 3 orders of white light interference. The OPD of a biological cell is typically smaller than $\lambda_0$ which allows quantitative reconstructions even if the interference-order of the background is shifted by one lambda.

The inherent appearance of ghost images (e.g. false-colors visualized in Fig 4 as a red arrow) resulting from the inserted sample at a distance of about half the FOV away, limits the interpretability of the JL-images for users and the algorithms. In general this also limits the technique to sparse phase-samples only. To avoid this effect for the calibration object we choose a reference sample which covered about 1/4 of the field of view (FOV). This is small enough to avoid any superposition of the ghost image with the actual interference of the object and large enough to ensure a high resolution for inverting the LUT. Additionally the glass-fiber was rotated so that the actual image was not overlaying the ghost image. The resulting JL-image as well as the known phase-variation of the fiber as a projection along the optical axis is visualized in Fig 3a). Here the left part *1* describes the raw JL image, the middle part *2* the computed OPD from the fiber and the right part *3* simulates the ideal RGB image using the forward model for the white-light interference.

To compare the results from the retrieved quantitative phase, we acquired ground-truth data of the same samples using the coherence-controlled holographic microscope (Q-Phase, Tescan and TELIGHT, Brno, Czech Republic)) equipped with an 40×, $NA_{det}$ = 0.95 lens for the detection shown in Fig 3. The maximum OPD of the glass-fiber is given in Fig 3b) which is used to scale the gray-scaled phase-images from the phase-retrieval algorithms.

### Results from image and video reconstruction

Fig 4c) shows an exemplary result after applying the direct pixel-wise L2 normalization. The algorithm has difficulties with phase-ambiguities due to near-by color-components indicated by a yellow arrow in the parametric plot of the RGB-color relationship for a given phase-sample in b). This effect is clearly visible in the zoomed-in region of interest (ROI) in d), where the missing relationship of the recovered phase-value of a single pixel to its surrounding pixels follows in a black rim around the cell.

Since the TV-based reconstruction algorithm preserves a relationship of nearby pixels by enforcing a piecewise constancy, it removes these artefacts (e.g. phase ambiguities due to nearby RGB values for different OPDs) visualized in the zoomed in ROI in Fig 4h). Additionally, due to the finite number of supporting points in the computed LUT mentioned previously, the L2-based algorithm follows in a stair-step-like reconstruction of the OPD which can is reduced by applying the TV-based algorithm.

By choosing the parameters $\lambda_{TV}$ and $eps_c$, we can control the shape of the result in terms of smoothness or piecewise constancy. We chose these parameters to be $\lambda_{TV} = 1.0$ and $eps_c = 0.001$ empirically but can be adjusted for different SNRs of the measurement. Nevertheless, the algorithm did not solve the problem in areas, where the ghost-image produced false-colors which are interpreted wrongly (Fig 4c), 4e) and 4f), red arrow). The retrieved phase-values agree with the ground-truth data from the CCHM measurements in Fig 3c). The reduced optical resolution in the JL images (e.g. hardly resolved nucleus) is caused by the reduced numerical aperture of the used lens (30×, $NA = 0.65$).

The open-sourced algorithm [26] was implemented using Tensorflows auto-differentiation mechanism. After initializing the iterative phase-recovery algorithm with the minimal-norm solution Eq 3 it typically converges within 50-100 iterations using the ADAM [36] update-rule and a learning-rate of $lr = 100$ thus taking ca. $5 - 10\ s$ on an ordinary laptop (CPU Intel i5, 8GB RAM) at $1024 \times 1024$ FOV. In case of video-data, the routine processes each camera-frame individually. To enable on-device reconstruction we also provide a web-based version of the algorithm [37] using the cloud-based image processing framework ImJoy [38].

In the supplemental Video (S1 Video,) we show reconstruction of a video file from a rotifer freely moving in filtered pont water. Here the reference fiber and sample of interest had to be captured using the same video-settings (e.g. framerate, color-map, bitrate, etc.) and background-color. The LUT-inversion is separately done for only one frame from the reference video. During our studies we found out, that a matching background color is one of the most important factors to produce good-quality reconstruction results.

## Validation of reconstructed data

In cases where the proposed image processing algorithm is applied to phase-objects with rather unknown phase-profile (e.g. biological cells) the accuracy of the reconstructed result is hard to estimate. To give an additional measure for the accuracy of the computed result to the comparison of holographic measurements above, we compare acquisitions of microspheres immersed in oil with a known phase-profile.

To show the error over time and across multiple frames, we use a series of video-frames extracted from a MP4 movie-stream acquired using a Huawei P20 cellphone (Shenzhen, China) by relying on the open-source codec FFMPEG [39]. The reference sample (e.g. optical fiber) was acquired using the same camera-settings and equal background of the white-light interference. As a known phase sample we use PMMA microspheres ($D = 1 - 20\ \mu m$, $n_{sphere} = 1.592$, micro particles GmbH, Germany) which were applied to a standard coverslip and diluted in custom-mixed immersion oil with known refractive index ($n_{oil} = 1.5321$, Cargille Labs, Series A). The resulting maximum OPD computes as $OPD = 2r_{sphere} \cdot (n_{sphere} - n_{oil})$ and can be used to create ground-truth data for the JL-measurements.

To give a pixel-to-pixel estimate for the resulting error, we create the 2D projection of a perfect 3D microsphere based on the proposed parameters and place them on the centre-coordinates of a sphere in each measurement. Parameters like centre-coordinates and radii were derived from a Hough-Transformation to have a measure of the error for many frames and spheres (exemplary visualised in Fig 5 and S2 Video.

During the measurements, we noticed, that since the maximum OPD of the spheres is more than one-$\lambda_c$ and the color sensitivity of the cellphone camera is less optimal (e.g. noise and the unavoidable auto-white balance in the video mode), the algorithm has difficulties to recover the quantitative phase of the spheroids correctly. Due to the fact, that we use a cylindrical object as a calibration target the number of measurement points per OPD-class is not equally distributed, which can be an additional factor for artefacts in the reconstruction in

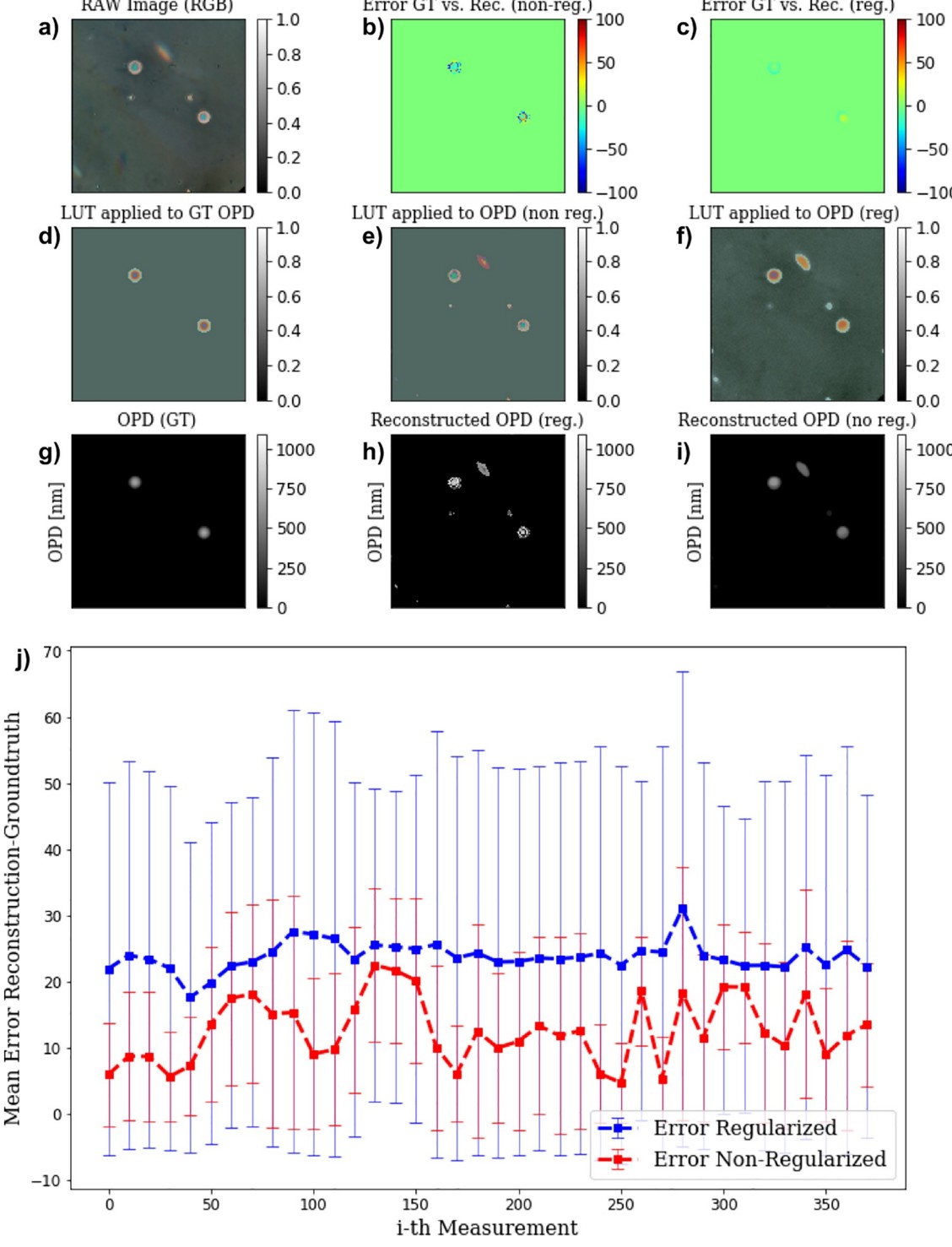

**Fig 5. Comparison of expected ground-truth and reconstructed OPD results.** The images show an exemplary frame from a video-sequence ().

cases where the samples have an OPD of more than one-$\lambda_c$. This recalls the need for a proper calibration target (e.g. phase wedge). We partially solved this with a adjusting the spacial sampling of the OPD in the LUT-creation (e.g. linear spacing between pixel-steps rather than OPD-steps) as described earlier.

In order to make sure, that the results are not suffering from a wrapped phase, the TV regularisation parameter has to be tuned more carefully. On the one hand an over-regularized reconstruction results in a smooth OPD solution with lower and underestimates the absolute OPD. On the other hand follows low regularisation in many phase-jumps. We tuned the parameter for the TV-regularisation to meet a balance between over- (e.g. too blurry results) and under-regularized results (e.g. too much phase-wrapping). Additionally we increased the number of iteration (e.g. 400 steps) to make sure that the algorithm converges.

To estimate the error between the expected ground-truth and the regularized/non-regularized reconstruction we compute the pixel-wise percentage error as $Err = (OPD_{GT} - OPD_{Reconstructed})/OPD_{max}$ (Fig 5b and 5c), where $OPD_{max}$ is the max OPD given by the reference fibers' profile. The graph in Fig 5j) shows the absolute mean error in percent and standard-deviation for each frame of the video-reconstruction (shown in), where the error is only computed in the masked error of known OPD (i.e. detected microspheres). The TV-regularized result has a much lower mean-error compared to the minimal norm solution but is less robust in situation, where the sample is moving and follows in a motion-blur artefact. In general, our proposed algorithm deals with the wrapped phase much better and produces results with a smaller mean-error and standard-deviation per frame as shown in graph j) in Fig 5. The mean-error for the whole time-series is given by 23.78 ± 2.15 for the minimal-norm solution and 12.58 ± 4.90 for the TV-regularized result.

All results and their corresponding algorithms can be reviewed in our github-repository ([26]).

## Discussion

We successfully showed, that by combining methods like the Jamin-Lebedeff interference-based quantitative phase microscope—sometimes considered as useless compared to state of-the-art fluorescent or holographic microscopes—with modern cellphones and 3D printed components, helped the device to a new live. Even more, together with tailored image-processing algorithms, we were able to recover information which was not possible using hitherto existing analogue methods. Jamin-Lebedeff microscopes are not widely spread, which makes accessing our method difficult. Yet, we hope, that our method inspires others to reuse historical lab-equipment carrying huge potential, if combined with modern imaging devices and image-processing algorithms.

The color-coded phase-information enabled us to gain quantitative OPD information in a single exposure, making it advantageous over sequential methods like Differential Phase Contrast (DPC) [33] or Fourier Ptychogarphy Microscopy (FPM) [33, 34] e.g. DPC or FPM e.g. for live-cell imaging where dynamic intra-cellular processes are subject of the observation. Even though industry-grade RGB CMOS cameras are not widely spread, we would like to emphasize, that the method is also applicable to those devices.

One main advantage of the proposed method with the generated LUT from a reference sample is, that it does not rely on specific light-source spectrum or a calibrated camera. Once the reference sample is recorded, many other phase-objects can be acquired under the same experimental conditions. During the reconstruction, we noticed that the method is very robust even in cases where the background was not set to black. A next step could be to detect areas

where false-colors e.g. produced by dust in the imaging path or superposing ghost-images cause the algorithm to produce false results. This could lead to a probability how likely the reconstructed matches the expected result in a given experiment. Alternatively this could be solved by introducing an additional mask which forces sample-free regions to be the background-phase. It can also be beneficial to use a wedge-like phase object which introduces a linear phase-ramp and produces better sampling for generating the LUT. Compared to optical fibers, this is rather hard to get with accurate dimensions.

In the recorded video, each frame is processed individually which unfortunately does not preserve the intra-frame phase relationship. This could be accounted for by applying a 3D ($x$, $y$, $t$) TV constrain or using machine-learning approaches which is capable of learning sequential-type data like Long-Short-Term-Memory (LSTM) neural networks or an adapted cost-function which accounts for multiple frames.

In the future we are planning to replace the iterative gradient-based phase-recovery with a trained neural network to further accelerate the phase-recovery. The crucial part is to create a good training dataset with data pairs corresponding to a real-world experiment. With our algorithm this process can be automated by recording a large set of microscopic phase objects and using this for training e.g. a conditional generative adversarial network (cGAN) [34, 35] and directly deploying it on the cellphone.

Another point is that our computational method might be useful not only in biological interference microscopy but also in the polarizing microscopy of minerals. The origin of the phase retardation is different, in that it results from birefringence, but there is a similar problem in that mineral specimens are conventionally cut at 30 micrometers thickness and the birefringence is then be deduced from retardation in length units divided by the known thickness. Mineralogists learn to distinguish the retardation colors of the different orders by eye, but our method might remove ambiguity and the need for skilled observation as well as providing a more accurate and objective estimate.

## Supporting information

**S1 Video. Comparison of different reconstruction algorithm in live-cell imaging of rotifer.** In this video we give a comparison of raw RGB data, the minimal-norm solution and the TV-regularized iterative algorithm of a live-cell image series with a framerate of 30 frames per second (fps).
(GIF)

**S2 Video. Validation of time-series reconstruction of spherical beads.** The graphs give the percentaged error between the expected ground-truth measurements and the reconstruction using regularized and non-regularized computations of the OPD.
(GIF)

## Acknowledgments

We thank especially Jan Peychl from the Light Microscopy Facility of the MPI-CBG in Dresden for giving us the opportunity to use the quantitative phase microscope (Q-Phase, Tescan and TELIGHT, Brno, Czech Republic). Special thanks goes to Patrick McCall and Moritz Kreysing who helped us acquiring the reference images using the Q-Phase at the MPI-CBG in Dresden. We also thank Patrick Then from the Leibniz IPHT in Jena for providing fixed cell-samples. We also thank Tobias Tiess and Tina Eschrich from IPHT for providing the fibre used as a reference sample.

## Author Contributions

**Conceptualization:** Benedict Diederich, Barbora Marsikova, Brad Amos, Rainer Heintzmann.

**Data curation:** Benedict Diederich.

**Formal analysis:** Benedict Diederich, Barbora Marsikova, Rainer Heintzmann.

**Funding acquisition:** Benedict Diederich, Brad Amos, Rainer Heintzmann.

**Investigation:** Benedict Diederich, Barbora Marsikova, Rainer Heintzmann.

**Methodology:** Benedict Diederich, Brad Amos, Rainer Heintzmann.

**Project administration:** Benedict Diederich, Rainer Heintzmann.

**Resources:** Benedict Diederich, Barbora Marsikova, Brad Amos, Rainer Heintzmann.

**Software:** Benedict Diederich, Rainer Heintzmann.

**Supervision:** Benedict Diederich, Rainer Heintzmann.

**Validation:** Benedict Diederich, Barbora Marsikova, Brad Amos, Rainer Heintzmann.

**Visualization:** Benedict Diederich, Barbora Marsikova, Brad Amos, Rainer Heintzmann.

**Writing – original draft:** Benedict Diederich, Barbora Marsikova.

**Writing – review & editing:** Benedict Diederich, Barbora Marsikova, Brad Amos, Rainer Heintzmann.

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
