## [Decision Letter · Decision Letter 0]

24 Oct 2019

PONE-D-19-26395

One-Shot phase-recovery using a Cellphone RGB Camera on a Jamin-Lebedeff Microscope

PLOS ONE

Dear Mr. Diederich,

Thank you for submitting your manuscript to PLOS ONE. After careful consideration, we feel that it has merit but requires major revisions to meet PLOS ONE’s publication criteria. Therefore, we invite you to submit a revised version of the manuscript that addresses the points raised during the review process.

The revisions relates to:

1. Further phase measurements which are quantified and compared with respect to 'ground truth' phase object,

2. Quantitative analysis of the temporal and spatial standard deviations of the obtained OPD map.

We would appreciate receiving your revised manuscript by Dec 08 2019 11:59PM. To enhance the reproducibility of your results, we recommend that if applicable you deposit your laboratory protocols in protocols.io, where a protocol can be assigned its own identifier (DOI) such that it can be cited independently in the future. For instructions see: http://journals.plos.org/plosone/s/submission-guidelines#loc-laboratory-protocols

We look forward to receiving your revised manuscript.

Kind regards,

Yuval Garini, Ph.D.

Academic Editor

PLOS ONE

**Journal Requirements:**

**Comments to the Author**

1. Is the manuscript technically sound, and do the data support the conclusions?

Reviewer #1: Partly

Reviewer #2: Yes

2. Has the statistical analysis been performed appropriately and rigorously? 

Reviewer #1: N/A

Reviewer #2: I Don't Know

3. Have the authors made all data underlying the findings in their manuscript fully available?

Reviewer #1: Yes

Reviewer #2: Yes

4. Is the manuscript presented in an intelligible fashion and written in standard English?

Reviewer #1: Yes

Reviewer #2: Yes

5. Review Comments to the Author

Reviewer #1: The authors suggest using Jamin-Lebedeff (JL) polarization interference microscopy for quantitative phase microscopy of biological cells. The optical path difference (OPD) between the sample and the reference, which should be empty, is encoded into a colour by white-light interference. Then, a colour-table is used to calculate the OPD map. The paper uses a phase retrieval method relying on a single-shot measurement, allowing real-time quantitative phase imaging. The system includes a cellphone with an RGB camera. The problem in the paper is that there are not enough experimental results, which are quantified in respect to a ground truth. Cells might come in different shapes and heights. I think you should add imaging of known diameter microscopic beads (5-10 micron), and also add a quantitative analysis that gives the values of the temporal and spatial standard deviations of the obtained OPD map. Finally, the video does not run. Please submit in a different format.

Reviewer #2: To meet the need of quantitative phase measurements, a cellphone RGB camera is used to capture image from a Jamin-Lebedeff microscope and look-up-table (LUT) is created to convert the measured RGB signal of a phase-sample into an OPD. The method is straightforward, but the accuracy of phase measurement is uncertain. While the smart phone can compute the phase directly, it needs extra effort to get a good images than using the regular microscope camera directly. The contribution to demonstrated application is very limited.

6. PLOS authors have the option to publish the peer review history of their article (what does this mean?). If published, this will include your full peer review and any attached files.

Reviewer #1: No

Reviewer #2: No

---

## [Author Response · Author response to Decision Letter 0]

10 Dec 2019

Dear editor, 

Thank you very much for summarizing the issues aroused by the reviewers. We did a generous analysis of long-time image acquisition with ground-truth data which can be found in the manuscript and in the newly attached Supplementary video. 

We give a step-by-step explanation of all improvements in the rebuttal letter. 

If you have any questions, please don't hesitate to get back to us at any time. 

Kind regards

Benedict Diederich

---

## [Editor Report · Decision Letter 1]

13 Dec 2019

One-Shot phase-recovery using a Cellphone RGB Camera on a Jamin-Lebedeff Microscope

PONE-D-19-26395R1

Dear Dr. Diederich,

Thank you for sending the revised manuscript. It answers all the issues raised by the authors and has been judged scientifically suitable for publication. It is therefore formally accepted for publication once it complies with all outstanding technical requirements.

Note however the request for minor changes below.

With kind regards,

Yuval Garini, Ph.D.

Academic Editor

PLOS ONE

MINOR CHANGES: 

Please go over the manuscript to clarify minor editing issues such as:

1. Page 6, line 198 - it points to Fig 4 (blue) - is it Fig 4a? Please add the exact figure.

2. Page 6 lines 199-200 - Which figure is it? Fig 4b?

3. Same place, I think it should be a yellow arrow. Also in the figure caption.

4. Ensure that all figure references are exact (i.e. Fig 4b).

5. Page 8, line 288: what is (4, red circle)? If it is Figure 4, there are no circles in figure 4.

6. Page 10, line 353: The Err is defined. Accordingly, the value should be a number smaller than 1.

    But the values given are large. Explain

7. The links in ref's 26, 31 does not work.

8. Page 8, line 261: What is 3? Fig. 3?

---

## [Editor Report · Acceptance letter]

18 Dec 2019

PONE-D-19-26395R1 

One-Shot phase-recovery using a Cellphone RGB Camera on a Jamin-Lebedeff Microscope 

Dear Dr. Diederich:

I am pleased to inform you that your manuscript has been deemed suitable for publication in PLOS ONE. Congratulations! Your manuscript is now with our production department. 

With kind regards,

on behalf of

Prof. Yuval Garini 

Academic Editor

PLOS ONE